# African Swine Fever Virus pE199L Induces Mitochondrial-Dependent Apoptosis

**DOI:** 10.3390/v13112240

**Published:** 2021-11-08

**Authors:** Tingting Li, Gaihong Zhao, Taoqing Zhang, Zhaoxia Zhang, Xin Chen, Jie Song, Xiao Wang, Jiangnan Li, Li Huang, Lili Wen, Changyao Li, Dongming Zhao, Xijun He, Zhigao Bu, Jun Zheng, Changjiang Weng

**Affiliations:** Division of Fundamental Immunology, National African Swine Fever Para-Reference Laboratory, State Key Laboratory of Veterinary Biotechnology, Harbin Veterinary Research Institute, Chinese Academy of Agricultural Sciences (CAAS), Harbin 150069, China; sunshine8302021@163.com (T.L.); 13359317177@163.com (G.Z.); 18804603005@163.com (T.Z.); zzx_ab@163.com (Z.Z.); Chenxin2021202109@163.com (X.C.); jonepiecej@163.com (J.S.); 13220632366@163.com (X.W.); lijiangnan521@163.com (J.L.); highlight0315@163.com (L.H.); W18811617029@163.com (L.W.); lichy0420@163.com (C.L.); zhaodongming@caas.cn (D.Z.); hexijun@caas.cn (X.H.); buzhigao@caas.cn (Z.B.)

**Keywords:** African swine fever virus, pE199L, mitochondria, apoptosis

## Abstract

African swine fever (ASF) is a severe hemorrhagic disease in swine characterized by massive lymphocyte depletion and cell death, with apoptosis and necrosis in infected lymphoid tissues. However, the molecular mechanism regarding ASFV-induced cell death remains largely unknown. In this study, 94 ASFV-encoded proteins were screened to determine the viral proteins involved in cell death in vitro, and pE199L showed the most significant effect. Ectopic expression of pE199L in porcine cells (CRL-2843) and human cells (HEK293T and HeLa cells) induced cell death remarkably, showing obvious shrinking, blistering, apoptotic bodies, and nuclear DNA fragments. Meanwhile, cell death was markedly alleviated when the expression of pE199L was knocked down during ASFV infection. Additionally, the expression of pE199L caused a loss of mitochondrial membrane potential, release of cytochrome C, and caspase-9 and -3/7 activation, indicating that the mitochondrial apoptotic pathway was involved in pE199L-induced apoptosis. Further investigations showed that pE199L interacted with several anti-apoptotic BCL-2 subfamily members (such as BCL-X_L_, MCL-1, BCL-W, and BCL-2A1) and competed with BAK for BCL-X_L_, which promoted BAK and BAX activation. Taken together, ASFV pE199L induces the mitochondrial-dependent apoptosis, which may provide clues for a comprehensive understanding of ASFV pathogenesis.

## 1. Introduction

African swine fever (ASF) is a highly contagious viral disease in swine characterized by hemorrhagic fever and high mortality rates in acute ASFV-infected herds. ASF is currently spreading in Europe, Russia, China, and other Asian countries, becoming a serious threat to global pork production [1]. The causative agent of ASF is African swine fever virus (ASFV) which contains a double-stranded DNA genome of about 170–193 kbp in length and encodes more than 150 proteins [2]. ASFV mainly replicates in porcine macrophages and monocytes and induces cell death in infected lymphoid tissues [3,4], which is supposedly responsible for its pathogenesis [5].

Apoptosis is a finely described form of cell death which plays crucial roles in removing virus-infected cells to maintain tissue homeostasis [6]. Apoptosis is mainly induced via two pathways: extrinsic and intrinsic. The extrinsic apoptotic pathway involves caspase-8 activation by recruiting the adaptor molecule Fas-associated death domain (FADD) upon Fas/TNF-ligand binding to FAS/TNF receptors. The intrinsic (also known as mitochondrial) apoptotic pathway mainly involves the regulation of the B cell lymphoma 2 (BCL-2) family members for mitochondrial outer membrane permeabilization (MOMP) [7]. The BCL-2 family is usually subdivided into three subgroups based on their structures and functions: anti-apoptotic members (such as BCL-2, BCL-W, BCL-X_L_, and MCL-1), pro-apoptotic effectors (prominently BAK and BAX), and pro-apoptotic BH3-only members (such as NOXA, BIM, PUMA, and BID) [8,9]. The specificity and affinity between the subgroup members vary, allowing for a complex interplay to determine variability in cell fate [8]. In healthy cells, anti-apoptotic members prevent MOMP by binding activated BAK and BAX effectors or BH3-only proteins. Following cytotoxic stimulation, several BH3-only proteins compete with BAX/BAK for anti-apoptotic BCL-2 family members to release and activate BAX/BAK [8,10,11,12]. Upon activation, BAK oligomerization occurs in the outer mitochondrial membrane, while BAX undergoes conformational changes and translocation to the mitochondria, resulting in MOMP [13]. Subsequently, the mitochondria releases apoptosis-related proteins such as cytochrome C (cyto C), which in turn activates the downstream caspase cascade (e.g., caspase-9 and caspase-3/-7), and finally initiates apoptosis [7].

Viruses have evolved diverse strategies to regulate apoptosis and ensure their propagation [14]. It has been reported that many viruses encode pro-apoptotic proteins, targeting the mitochondrial apoptotic pathway [15,16]. Severe acute respiratory syndrome coronavirus (SARS-CoV-1) encodes the 7A protein and induces apoptosis through binding BCL-X_L_ [17]. Additionally, many viruses also encode anti-apoptotic proteins, which may help to evade or delay the early onset of apoptosis at the early stage of viral infection [6,18]. Notably, ASFV encodes not only pro-apoptotic proteins (e.g., pE183L), but also anti-apoptotic proteins (e.g., pA179L, pA224L, pEP153R, and pDP71L) [19,20,21,22,23], suggesting that ASFV may require different apoptotic regulators at different stages of replication. For example, ASFV pE183L induces apoptosis by possibly binding the 8 kDa dynein light chain (DLC8) for BIM translocation to mitochondria [23,24]. Previous studies demonstrated that ASFV pA179L blocked apoptosis by interacting with pro-apoptosis effectors and BH3-only proteins [19,25]. Therefore, identification of ASFV-encoded pro-apoptotic proteins will advance our understanding about ASFV pathogenesis. 

However, due to a lack of information regarding the function of most ASFV-encoded proteins, the mechanism of ASFV pathogenesis remains largely unclear. Thus, we analyzed the effect of 94 ASFV genes on cell death in vitro and found several genes, including the *E199L* gene. It has been reported that the pE199L is expressed in the late stage of ASFV infection, as confirmed in various ASFV isolates [26]. ASFV pE199L is a multifunctional protein involved in virus entry [27], and promotes cell autophagy [28]. These reports suggest that ASFV pE199L is closely related to its pathogenesis. In this study, we found that expression of pE199L in HEK293T and HeLa cells could significantly induce mitochondrial-dependent apoptosis via competition with BAK for BCL-X_L_, which could lay the foundation for further studies on the mechanism of ASFV-induced cell death.

## 2. Materials and Methods

### 2.1. Facility and Animal Experiment

All the experiments with live ASFV were conducted within the enhanced biosafety level 3 (P3+) and level 4 (P4) facilities at the Harbin Veterinary Research Institute (HVRI) of the Chinese Academy of Agricultural Sciences (CAAS) and were approved by the Ministry of Agriculture and Rural Affairs and China National Accreditation Service for Conformity Assessment.

In total, 4 SPF pigs (7 weeks of age) were intramuscularly inoculated with HLJ/18 isolates at 10^3^ HAD_50_, and 1 pig was inoculated with PBS. The pigs were monitored daily for 12 days for temperature and mortality. Tissues and organs were collected from the dead pigs and the surviving pigs which were euthanized on day 12 post-challenge for viral DNA quantification by qRT-PCR. Gastric hepatic lymph nodes (GHLNs), thymuses, and tonsils from infected or non-infected pigs were stained with hematoxylin and eosin or TUNEL to detect cell death.

### 2.2. Cell Lines and Antibodies

Primary porcine alveolar macrophages (PAMs) were collected from 30-day-old SPF pigs and were cultured in RPMI 1640 supplemented with 10% fetal bovine serum (FBS), 100 U/mL penicillin, and 100 μg/mL streptomycin at 37 °C with 5% CO_2_. The PAM cell line 3D4/21 (CRL-2843), established by transformation of PAMs with SV40 large T antigen [29], was purchased from the American Type Culture Collection (ATCC) and cultured in RPMI-1640 medium supplemented with 10% FBS. HEK293T cells preserved in our lab were cultured in Dulbecco’s modified Eagle’s medium (DMEM) supplemented with 10% fetal bovine serum (FBS), 100 U/mL penicillin, and 100 μg/mL streptomycin at 37 °C with 5% CO_2_. HeLa-∆BAK and its parental cells were purchased from the EdiGene incorporated (catalog#: LS0032850802A, Beijing, China). 

The primary antibodies used in this study were specific for Tom20 (BD Biosciences, Franklin Lakes, NJ, USA), GAPDH (Proteintech, Nanjing, China), BAK (Abcam, Cambridge, Cambridgeshire, UK), BAX-NT (Proteintech, Nanjing, China), BAX6A7 (sc-23959, SANTA CRUZ, immunoprecipitation: 2 µg per 100 µg of total protein in 0.5 mL of cell lysate, Dallas, TX, USA), cytochrome C (Cell Signaling Technology, Boston, MA, USA), Cox IV (Cell Signaling Technology, Boston, MA, USA), Flag (Cell Signaling Technology, Boston, MA, USA), HA (Cell Signaling Technology, Boston, MA, USA), and GFP (Abcam, Cambridge, Cambridgeshire, UK). The anti-p72 mouse polyclonal antibody was prepared in our lab. The secondary antibodies used in Western blotting and IFA were for anti-mouse IgG (H+L) DyLight™ 800-Labeled (042-07-18-06) from Sera Care (Milford, MA, USA), IRDye^®^ 800CW goat anti-rabbit IgG (H+L) (925–32211) from LI-COR (Lincoln, NE, USA), and Alexa Fluor 594-conjugated goat anti-mouse IgG (H+L) F(ab=)2 fragment (Thermo Fisher Scientific, Sunnyvale, CA, USA).

### 2.3. Virus Titration

The ASFV HLJ/18 strain (ASFV HLJ/18) (GenBank accession: MK333180.1) was propagated in PAMs, and the virus titers were determined by the hemadsorption (HAD) assay. The HAD assay was performed as described previously, with minor modifications [30]. Briefly, primary peripheral blood monocytes (PBMs) (2 × 10^5^ cell per well) were seeded in 96-well plates for incubation over 24 h and subsequently infected with 20 μL per well of 10-fold serially diluted supernatant in quintuplicate. Then, 100 μL per well of 1% erythrocytes were added. The HAD was observed for 7 days, and 50% HAD doses (HAD_50_) were calculated by using the method of Reed and Muench [31]. 

Viral genomic DNA of ASFV-infected PAMs and culture supernatants was extracted by using QIAGEN DNA Mini Kits (QIAGEN, Germantown, MD, USA). The viral growth in the cells and supernatants were determined with viral p72 gene copy numbers by qRT-PCR. The qRT-PCR primers and probe used in this study are listed in Table 1. 

### 2.4. Construction of Recombinant Plasmids

According to the genomic sequence of ASFV HLJ/18, the cDNAs corresponding to 94 ASFV-encoded proteins were synthesized by GenScript incorporated (Nanjing, China) and cloned into the pCAGGS-Flag vector. The cDNAs corresponding to BCL-2 family members (BAK, BAX, BCL-X_L_, MCL-1, BCL-W, BCL-2, and BCL-2A1) were cloned into the pCAGGS-HA or pCAGGS-Flag vector. The mutants of BCL-X_L_ were cloned into the pCAGGS-HA vector. The *E199L* or *E1**83L* genes were cloned into the pEGFP-C1 vector and respectively named pGFP-E199L and pGFP-E183L. All constructs were validated by DNA sequencing. The primers used in this study are listed in Table 1.

### 2.5. Cell Viability Assay

Cell viability was determined by the CellTiter-Glo Luminescent Cell Viability Assay from Promega incorporated (Madison, WI, USA) [32]. In brief, HEK293T cells were seeded at a density of 1 × 10^4^ cells per well in 96-well plates from Corning incorporated (New York, NY, USA) for 12 h and then transfected with the indicated plasmids for 36 h. CellTiter-Glo^®^ 2.0 Reagent was added to the cell cultures at a 1:1 volume dilution. The ATP activity was measured using an Enspire multiscan spectrum reader (Waltham, MA, USA).

### 2.6. Apoptosis Assay

According to the manufacturer’s instructions for the Annexin V-FITC/PI assay kit (BD Pharmingen^TM^, Franklin Lakes, NJ, USA), CRL-2843, HEK293T, HeLa, or HeLa-∆BAK cells were transfected with the indicated plasmids and then stained with Annexin V-FITC (0.1 g/mL) and propidium iodide (PI) (1 μg/mL) or PI alone. PAMs were infected with ASFV HLJ/18 at the indicated MOIs and stained with PI at different time points post infection (12, 24, and 36 hpi). Then, 10,000 cells were analyzed by a Cytomics FC 500 flow cytometer (Beckman Coulter incorporated, Brea, CA, USA).

### 2.7. Caspase Activity Assay

HeLa cells were transfected with the indicated plasmids for 36 h or PAMs were infected with 0.1, 1, or 10 MOI for 24 h. Cells were collected and detected using Caspase-Glo^®^ 8, 9, 3/7 Assay (Promega incorporated, Madison, WI, USA) according to the manufacturer’s instructions.

### 2.8. TUNEL Assay

According to the manufacturer’s instructions for the One Step TUNEL Apoptosis Assay kit (Beyotime incorporated, Beijing, China), ASFV-infected tissue sections or HEK293T cells transfected with indicated plasmids were fixed with 4% paraformaldehyde for 30 min, permeabilized with 0.3% Triton X-100 in 1× PBS, and then stained with TUNEL reagent for 1 h at 37 °C The images were examined by a Zeiss LSM-800 laser scanning fluorescence microscope (Carl Zeiss AG, Oberkochen, Germany). DNA fragments were assessed by counting the percentage of TUNEL-labeled cells in at least 100 GFP-expressing or GFP-pE199L expressing cells for each sample under a 63× oil objective. 

### 2.9. Measurement of the Mitochondrial Membrane Potential

To test the mitochondrial membrane potential with JC-1 [33], HeLa or HeLa-∆BAK cells were transfected with pFlag-E199L or empty vector for 36 h, or PAMs were infected with ASFV HLJ/18 at 1 MOI for the indicated times post infection. According to the manufacturer’s instructions for the MitoProbe Assay Kit (Invitrogen, Carlsbad, CA, USA), the cells were stained with JC-1 dye (2 nM) at 37 °C for 30 min and 10,000 cells were analyzed on a Cytomics FC 500 flow cytometer (Beckman Coulter incorporated, Brea, CA, USA), with appropriate excitation using emission at 529 nm (green) and 590 nm (red). The mitochondrial membrane potential is indicated by a ratio of red fluorescence intensity to green fluorescence intensity. 

### 2.10. RNA Extraction and Quantitative Reverse-Transcription Polymerase Chain Reaction (qRT-PCR)

Total RNA from ASFV-infected PAMs was extracted using TRIzol reagent (Invitrogen, Carlsbad, CA, USA), and reverse transcription was performed with a PrimeScript™ RT Reagent Kit (Takara, Kusatsu, Shiga, Japan). To detect ASFV *E199L* mRNA expression, reverse-transcription products were amplified using a QuantStudio 5 system (Applied Biosystems, Sunnyvale, CA, USA) with SYBR^®^ Premix ExTaq™ II (Takara, Kusatsu, Shiga, Japan) according to the manufacturer’s instructions. Data were normalized to the level of HPRT expression in each sample. The qRT-PCR primers used in this study are listed in Table 1.

### 2.11. Immunofluorescence Assay (IFA) and Real-Time Confocal Analysis

HeLa cells were transfected with the indicated plasmids for 36 h, and were fixed with 4% paraformaldehyde for 30 min at 4 °C and permeabilized with 0.3% Triton X-100 for 10 min, followed by blocking with 5% bovine serum albumin (BSA) for 1 h. The cells were incubated with mouse anti-Tom20 primary antibodies for 1 h and then stained with Alexa Fluor 594-labeled goat anti-mouse IgG for 1 h and DAPI for 10 min. The subcellular localization was visualized using a Zeiss LSM-800 laser scanning fluorescence microscope (Carl Zeiss AG, Oberkochen, Germany) under a 63× oil objective. 

To examine the cell morphology and mitochondrial dynamics, HeLa cells were transfected with the pGFP-E199L plasmid alone or together with pDsRed-mito (used as a mitochondrial marker, BioVector NTCC incorporated, Beijing, China). Real-time confocal images were recorded every 5 min or 8 min using a Zeiss LSM-800 laser scanning fluorescence microscope.

### 2.12. Co-IP and Western Blotting Assay

Co-IP and Western blotting analyses were performed as previously described [34]. The detailed experimental procedures are as follows. For Co-IP, whole-cell extracts were lysed in lysis buffer (50 mM Tris-HCl, pH = 7.4, 150 mM NaCl, 5 mM MgCl_2_, 1 mM EDTA, 1% Triton X-100, and 10% glycerol) containing 1 mM PMSF and 1× protease inhibitor cocktail (Roche incorporated, Basel, Switzerland). Then, the cell lysates were incubated with anti-Flag (M2) agarose beads (Sigma, Saint Louis, MO, USA) at 4 °C for 8 h on a roller. The precipitated beads were washed 5 times with cell lysis buffer. For Western blot analysis, equal amounts of cell lysates and immunoprecipitants were resolved by 12–15% SDS-PAGE gel and then transferred to a PVDF membrane (Millipore incorporated, Billerica, MA, USA). After incubation with primary and secondary antibodies, the membranes were visualized by an Odyssey 2-color infrared fluorescence imaging system (LI-COR, Lincoln, NE, USA).

### 2.13. Mitochondria Purification, Cytochrome C Release, BAK Cross-Linking, and BAX Activation 

Mitochondria were prepared using Minute^TM^ Mitochondria Isolation Kit (MP007, Invent Biotechnologies, Eden Prairie, MN, USA) according to the manufacturer’s instructions. Briefly, HeLa cells were seeded on a culture dish (10 cm), incubated overnight, and then transfected with pCAGGS-Flag or pFlag-E199L for 36 h, or treated with staurosporine (30 nM) for 12 h as a positive control. The cells (3 × 10^7^) were collected and subjected to mitochondrial purification. Meanwhile, several cells (2 × 10^6^) were lysed as input in the BAK cross-linking experiment.

For the release of cytochrome C from mitochondria, cytosol and mitochondrial fractions were separated and equal amounts of proteins from each fraction were immunoblotted with anti-cytochrome C and anti-Flag. The Cox IV and GAPDH were respectively used as the internal control of the mitochondrial or cytosolic fraction. 

For BAK cross-linking, pelleted mitochondria were resuspended in PBS (pH 7.2) and subjected to cross-linking using 0.2 mM bismaleimidohexane (BMH) for 30 min at room temperature. The reactions were centrifuged at 13,000× *g* and pellets were analyzed by Western blotting with an anti-BAK antibody (Sigma, Saint Louis, MO, USA).

For the detection of BAX translocation, cytosol and mitochondrial fractions were separated, and equal amounts of proteins from each fraction were immunoblotted with anti-BAX-NT, anti-Flag, anti-Cox IV, or anti-GAPDH. 

### 2.14. Small Interfering Rnas Assay

Small interfering RNAs (siRNAs) that target ASFV *E199L* or non-targeting siRNAs (siNC) were chemically synthesized (GenePharma incorporated, Shanghai, China). In brief, PAMs were transfected with the corresponding siRNA (90 pmol) using Lipofectamine RNAiMAX Transfection Reagent from Invitrogen (Carlsbad, CA, USA) according to the manufacturer’s instructions. At 12 h post-transfection, the cells were transfected with pFlag-E199L or infected with ASFV HLJ/18 at 1 MOI for 24 h. The cells were collected and analyzed by qRT-PCR and Western blotting, with analysis of apoptosis using PI staining by flow cytometry. The siRNAs used in this study are listed in Table 1.

### 2.15. Electron Microscopy Imaging

Transmission electron microscopy (TEM) observations of apoptosis was carried out as previously described [35]. HEK293T cells were transfected with the indicated plasmids for 36 h, and then fixed and embedded. Apoptotic cells were observed and quantified in per sample under a JEM2100 transmission electron microscope (JEOL, Tokyo, Japan). 

### 2.16. Statistical Analysis

All statistical analyses were performed using GraphPad Prism 8 software (GraphPad Software Inc., San Diego, CA, USA). The normal distribution of the data was assessed using the Shapiro–Wilk test and the D’Agostino and Pearson test. The standard deviations (SDs) of the data were compared using the F test or the Brown–Forsythe test and the Bartlett’s test. In the study, for the comparison of 2 groups, normally distributed data were analyzed using the Student’s unpaired *t*-test (2-tailed parametric test), while normally distributed data with unequal SDs were analyzed using the unpaired *t*-test with Welch’s test (2-tailed parametric test, Figure 1D). For multiple comparisons, normally distributed data were analyzed using the ordinary ANOVA test with Dunnett or the Brown–Forsythe and Welch ANOVA test with Dunnett T3 (unequal SDs, Appendix A). For grouped analyses, 2-way ANOVA with the Sidak or Tukey test (Figure 1G) was applied for grouped analyses. Not normally distributed data were analyzed with the Mann–Whitney test (nonparametric test, Figure 1E and Appendix A). Data are presented as the mean ± standard deviation (SD), *n* = 3 unless otherwise stated. *P* values of less than 0.05 were considered statistically significant (* < 0.05).

## 3. Results

### 3.1. ASFV HLJ/18 Infection Induces Tissue Injury and Cell Death

To investigate the pathogenicity of the ASFV HLJ/18, 4 specific-pathogen-free (SPF) pigs (7 weeks of age) were intramuscularly injected with 10^3^ HAD_50_ of the ASFV HLJ/18 strain, and 1 pig was treated with PBS as control. As shown in Figure 1A,B, all pigs inoculated with ASFV HLJ/18 developed fever from 4 days post-infection (dpi) onwards and died within 11 dpi. Higher copies of viral genomic DNA were detected in the organs and tissues from ASFV-infected piglets (Figure 1C), and more severe cell death (arrowhead-indicated) and apoptotic cells (TUNEL-labeled, red) were observed in gastric hepatic lymph nodes (GHLNs), thymuses, and tonsils from ASFV-infected piglets as compared to the control (non-infection group) (Figure 1D). Furthermore, as shown in Figure 1E, the primary porcine alveolar macrophages (PAMs) infected with ASFV HLJ/18 became cytoplasmically vacuolized, rounded, and reduced in size, displaying more severe cytopathic effects (CPEs) compared with the control. The increased CPEs were correlated with viral growth, as shown by the increased expression of p72. To further analyze the cell death associated with ASFV-induced CPEs in PAMs, the cells were stained with propidium iodide (PI) and then analyzed by flow cytometry. The results showed that the number of PI-labeled cells significantly increased at 24 h post infection (hpi) as compared with the control (Figure 1F,G). These results demonstrate that ASFV HLJ/18 strain is a virulent isolate and can induce cell death in vivo and in vitro.

### 3.2. ASFV pE199L Induces Cell Death 

Previous reports showed that ASFV infection influenced cell apoptosis [4]. Studies have reported that pA179L and pA224L serve as anti-apoptosis-related proteins, while pE183L is a pro-apoptotic protein [19,20,23]. To identify more ASFV-encoded proteins involved in ASFV-induced cell death, 94 ASFV-encoded proteins were respectively expressed in HEK293T cells, and then the cell viability was detected by the CellTiter-Glo^TM^ Luminescent Assay. As shown in Figure 2A, the expression of pE199L, pB117L, pCP123L, and pE183L dramatically impacted cell viability. ASFV pE183L, a known apoptosis inducer, was identified, indicating the assay was reliable (Figure 2A). Notably, pE199L showed the most significant effect, and thus it was investigated in greater depth. Further verification showed that cell viability in the GFP-pE199L or Flag-pE199L-expressing group significantly decreased as compared with the control group, and there was no apparent discrepancy between pE199L-expressing groups (Figure 2B). In addition, a PI staining assay was used to detect the cell death induced by Flag-pE199L and GFP-pE199L. The results showed that the percentage of PI-labeled cells in the Flag-pE199L- or Flag-pE183L-expressing groups was markedly higher as compared to the Flag group (Figure 2C). Similarly, ectopic expression of GFP-pE199L increased the percentage of PI-labeled cells as compared to the GFP group (Figure 2D). Fluorescence images showed more PI-stained cells in GFP-pE199L or GFP-pE183L-expressing cells as compared to GFP-expressing cells (Appendix A). We also found that ectopic expression of pE199L in the porcine alveolar macrophages cell line 3D4/21(CRL-2843) increased the percentage of PI-labeled cells (Figure 2E and Appendix A). These results indicate that ASFV pE199L can induce cell death in vitro.

To test pE199L expression kinetics in ASFV-infected PAMs, the mRNA level of *E199L* gene was detected at different timepoints. From 12 hpi onwards, the mRNA levels of *E199L* gene dramatically increased (Figure 2F), indicating that *E199L* was a viral late gene. Subsequently, we explored the effect of pE199L on ASFV-induced cell death. The siRNA interfering assay showed that expression of pE199L decreased remarkably in siE199L-transfected groups at mRNA and protein levels (Figure 2G,H), which suggested that these siE199L were effective. Furthermore, the percentage of cell death in the siE199L group was much lower than that of the siNC group (Figure 2I), which indicated that the pE199L was involved in ASFV-induced cell death. 

### 3.3. ASFV pE199L Induces Apoptosis

There are at least three types of cell death, including apoptosis, necrosis, and pyroptosis [36]. To further determine the type of cell death induced by pE199L, cell morphological changes, nuclear DNA fragments, and phosphatidylserine (PS) eversion on the cell membrane were detected by real-time confocal microscopy, TUNEL labeling, and Annexin V-FITC/PI staining, respectively. We observed that GFP-pE199L was distributed uniformly in the cytoplasm and then aggregated (Figure 3A and Appendix A). Along with this process, the aggregation of GFP-pE199L was accompanied by shrinking, blistering, and subsequent formation of vesicles in cells (characteristics of apoptosis). Consistent with the result, HEK293T cells with the aggregated GFP-pE199L could be labeled by the TUNEL assay (Figure 3B), and the percentage of TUNEL-labeled cells in the GFP-pE199L-expressing HEK293T cells was significantly higher than that of the GFP-expressing HEK293T cells (Figure 3C). The Annexin V/PI staining result showed that the apoptotic percentage of Flag-pE199L-expressing HEK293T cells increased dramatically compared with that of the cells transfected with pCAGGS-Flag (Flag) (Figure 3D,E). Hyperchromatic nuclear and karyopyknosis (red arrowhead) and apoptotic bodies (yellow asterisk) were observed in GFP-pE199L or Flag-pE199L expressing HEK293T cells under transmission electron microscopy (Figure 3F). Statistical analysis of Figure 3F showed that pE199L expression increased apoptosis markedly more than the control (Figure 3G). Collectively, these results suggest that ASFV pE199L induces apoptosis in vitro.

### 3.4. ASFV pE199L Triggers the Mitochondrial Apoptotic Pathway

Membrane-rich organelles, such as mitochondria and endoplasmic reticulum (ER), have been widely reported to be involved in the regulation of apoptosis [37,38]. However, there was no obvious co-localization between pE199L and mitochondria or ER makers (data not shown). Interestingly, we observed that the mitochondria in the GFP-pE199L-expressing cells displayed excessive fission in comparison with the GFP-expressing cells (Figure 4A and Appendix A). To further test whether pE199L affected the morphology of mitochondria, a real-time confocal assay was conducted to monitor the dynamics of mitochondria. As shown in Figure 4B and Appendix A, the mitochondria were elongated and divided to form network along the cytoplasm at the beginning, and then accumulated to form collapsed mitochondria in the GFP-pE199L-expressing cells. Simultaneous shrinking, blistering, and ultimately cell death (red arrowhead) were observed. Furthermore, the result showed that the mitochondrial membrane potential of the Flag-pE199L-expressing cells was significantly lower than that of control cells (Figure 4C and Appendix A), Meanwhile, the protein level of cyto C, a mitochondrial apoptosis-related protein, decreased in the mitochondrial fraction and concomitantly increased dramatically in cytosol of Flag-pE199L-expressing cells, suggesting that cyto C was released from the collapsed mitochondria (Figure 4D). Additionally, caspase-9 and caspase-3/7 in pE199L-expressing HeLa cells were markedly activated as compared with the control group (Figure 4E,F). These results indicate that pE199L induces apoptosis in a mitochondria-dependent manner.

### 3.5. ASFV pE199L Promotes BAK Activation by Competing with BAK for BCL-X_L_

Previous studies reported that MOMP was driven by effector pro-apoptotic members of the BCL-2 family (prominently BAK and BAX) [8,9]. Thus, we tested whether ASFV pE199L interacted with BAK or BAX. As shown in Figure 5A, ASFV pE199L did not directly interact with BAK or BAX. However, we found that pE199L widely interacted with anti-apoptotic proteins, including BCL-X_L_, MCL-1, BCL-W, and BCL-2A1, but not BCL-2 (Figure 5B). Furthermore, we proposed that pE199L might competitively disrupt the interaction between anti-apoptotic proteins and BAK/BAX. As shown in Figure 5C, the amount of pE199L bound to BCL-X_L_ increased with the increase in pE199L expression, while the amount of BAK bound to BCL-X_L_ decreased, indicating that pE199L competed with BAK for BCL-X_L_. Interactions between other anti-apoptotic proteins (BCL-X_L_, BCL-2A1, MCL-1, and BCL-W) and BAK/BAX were found not to be disrupted by pE199L (data not shown). Furthermore, we found that pE199L interacted with the region containing the BH3 domain of BCL-X_L_ (Figure 5D). These results indicated that pE199L might target BCL-X_L_ to activate BAK.

BAK activation involves a key event, BAK homo-oligomerization [39]. As shown in Figure 5E, BAK was mainly observed as intramolecularly linked monomers in the Flag group, while BAK oligomers were observed in the pE199L-expressing group and staurosporine-treated group, indicating that pE199L induced BAK homo-oligomerization. Furthermore, the result showed that the mitochondrial membrane potential in Flag-pE199L-expressing HeLa-ΔBAK cells was higher than that of the Flag-pE199L-expressing HeLa cells (Figure 5F). In addition, the percentage of PI-labeled cells in HeLa-ΔBAK cells expressing Flag-pE199L dramatically decreased compared with that of HeLa cells expressing Flag-pE199L (Figure 5G). We also found that pE199L could promote BAX translocation from the cytoplasm to the mitochondria (Figure 5H) and BAX activation via co-immunoprecipitation analysis with the anti-BAX6A7 antibody [40] (Figure 5I). Overall, our results indicate that the pE199L can disrupt the interaction of BCL-X_L_ and BAK, resulting in BAK homo-oligomerization and loss of the mitochondrial membrane potential.

### 3.6. Apoptosis Induced by ASFV Infection Contributes to Viral Growth 

To explore whether ASFV infection could induce the mitochondrial disruption, PAMs were infected with ASFV HLJ/18 at 1 MOI and were analyzed at 2, 6, 12, and 24 hpi. The remarkable loss of the mitochondrial membrane potential at 12 hpi onwards was observed compared to the control (non-infection) (Figure 6A and Appendix A). In addition, we noted that caspase-9 and the effector caspase-3/7 as well as caspase-8 were activated during ASFV infection (Figure 6B), suggesting that the intrinsic apoptotic pathway and the extrinsic apoptotic pathway were involved in ASFV infection. To further explore the effect of apoptosis on ASFV replication, PAMs were pretreated with z-DEVD-FMK, which inhibited activation of caspase-3 (an executioner caspase), and then infected with ASFV at 1 MOI. The viral replication efficacy was analyzed at 18 and 28 hpi. As shown in Appendix A, there was no significant difference in cell viability between the z-DEVD-FMK-treated group and the DMSO or control group. Furthermore, the results showed that z-DEVD-FMK markedly reduced ASFV genome copies in both cell cultures and supernatants compared to that of the DMSO group at 28 hpi (Figure 6C,D). Consistent with these results, the viral titers in supernatants were lower than those of the DMSO group (Figure 6E). Collectively, our findings showed that ASFV-induced apoptosis contributes to viral growth.

## 4. Discussion

In recent years, the unprecedented spread of ASF has caused substantial economic losses to the worldwide swine industry. Accumulating evidence has shown that ASFV infection induces cell death by apoptosis and necrosis, which are involved in viral pathogenicity by possibly promoting virus transmission [4,41,42,43]. We also observed apoptosis in lymphoid tissues when infected with ASFV HLJ/18. However, the underlying mechanism remains largely unknown. In this study, we found that (i) pE199L significantly induced cell death based on our screening in vitro and (ii) it could compete with BAK for BCL-X_L_, finally resulting in the mitochondrial-dependent apoptosis. Taken together, our findings provide a new clue for understanding ASFV pathogenesis.

Previous reports showed that various viral proteins initiated the programmed cell death in different manners during virus infection [16,17,44,45]. In the study, four cell death-related ASFV proteins were identified, including pE199L and pE183L. ASFV pE183L/p54 has been reported to cause apoptosis via the mitochondrial apoptotic pathway in Vero cells [23]. Recently, CD2v was reported to induced apoptosis by NF-κB in swine peripheral blood mononuclear cells [46]. However, CD2v was not identified in our screening system because released CD2v may require ligands such as CD58 to activate downstream death programs [46]. ASFV pE199L was identified as having the most significant effect in our screening system. Our results showed that ectopic expression of Flag-pE199L or GFP-pE199L in HEK293T cells markedly decreased cell viability, and the cell death induced by the two proteins did not reveal significant differences when analyzed via cellular ATP levels [32], suggesting that the fused-tag did not affect the function of pE199L on cell death. Additionally, we noted that ectopic expression of pE199L induced cell death in the HEK293T, HeLa, and porcine alveolar macrophage (CRL-2843) cell lines, suggesting that pE199L-induced cell death is a common phenomenon rather than being cell-type specific. Meanwhile, pE199L-mediated cell death showed obvious apoptotic features, including phosphatidylserine (PS) eversion, nuclear condensation and fragmentation, and apoptotic bodies. Furthermore, the knockdown of pE199L alleviated cell death to some extent during ASFV infection, indicating that pE199L behaved as a cell death inducer. We also noted that ectopic expression of pE199L in HEK293T induced 30–40% of cell death while ASFV infection at 1 MOI for 24 h in PAMs only induced 15% of cell death. We speculated that ASFV-encoding anti-apoptotic proteins [47] might inhibit apoptosis induced by pE199L. These reports and our findings indicate that ASFV-induced apoptosis is likely to involve multifaceted regulators containing pE199L. 

Mitochondria have been thought to be involved in virus-induced apoptotic responses and function as a pivotal organelle regulating apoptosis [48]. Hepatitis C virus-encoded NS4A protein alters the mitochondrial intracellular distribution, showing irregular-clumping and an aggregated form during apoptosis [16]. This phenomenon was also observed in ASFV-infected Vero cells [42], suggesting that ASFV might cause mitochondrial damage. Consistent with these results, ASFV HLJ/18 infection and pE199L expression could induce the mitochondrial collapse with the loss of the mitochondrial membrane potential and excessive fragmentation, suggesting that MOMP occurred. It has been reported that released cyto C from mitochondria binds to apoptotic peptidase-activating factor 1 (Apaf1), which recruits and activates the initiator caspase-9. Subsequently, the executioner caspase-3/7 is cleaved and activated [49]. Our study found that ectopic expression of pE199L induced the release of cyto C and the activation of caspase-9 and caspase-3/7. Meanwhile, we also found caspase-9 and caspase-3/7 activation during ASFV HLJ/18 infection. Although ASFV could activate the caspase-8 in PAMs, ectopic expression of pE199L did not induce caspase-8 activation (data not shown), implying that pE199L may not be involved in the extrinsic apoptotic pathway during ASFV infection. These observations indicate that pE199L activates the mitochondrial apoptotic pathway and imply that pE199L is closely associated with mitochondria damage during ASFV infection. 

Pro-apoptotic effectors, particularly BAK and BAX, play crucial roles in regulating apoptosis via the mitochondria-dependent pathway [8]. Schematic models for BAK and BAX activation have been proposed, in which anti-apoptotic BCL-2 family proteins (e.g., BCL-2, BCL-X_L_, and MCL-1) and BAK/BAX form a complex to block BAK/BAX activation. BH3-only proteins (such as BID, BIM, and PUMA) can directly disrupt the complex to activate BAK and BAX, resulting in apoptosis [12]. BAX and BAK are targeted by viruses to promote the progression of apoptosis, which seems to be an effective strategy for virus propagation [40,50]. The enterovirus 71-encoded 2B protein could directly interact with BAX and promote BAX activation [40]. It also has been reported that anti-apoptotic BCL-2 family proteins are involved in virus-mediated apoptosis [51,52]. Among them, BCL-X_L_ may usually serve as a target during viral infection due to its wider binding range to pro-apoptotic proteins than other known BCL-2 homologs [52,53]. Hepatitis B virus X protein targets BCL-X_L_ via its BH3-like motif to promote viral reproduction and infected-cell death [53]. In our study, pE199L widely interacted with several anti-apoptotic BCL-2 family members (such as BCL-X_L_ and MCL-1) instead of directly targeting BAK or BAX. Using the BCL-X_L_/BAK complex as a model, the results showed that pE199L competed with BAK for BCL-X_L_ which might depend on the region containing the BH3 domain of BCL-X_L_. Further investigation into the proposed model was not possible during ASFV infection due to the unavailable antibodies against endogenous pE199L. Notably, such a mechanism is also reported in SARS-CoV 7A protein and hepatitis B virus (HBV) X protein [17,44,53], suggesting that it is a general strategy for virus-induced apoptosis via the targeting of anti-apoptotic proteins. Currently, structural biology has been applied to explain the underlying mechanism responsible for apoptosis regulated by several viral proteins [54,55,56]. Thus, structural studies are still needed to investigate whether pE199L inserts into the groove formed by BH3, BH1, and BH2 of BCL-X_L_ to displace BAK, which may further support our results. Besides activated BAK, we found that BAX activation occurred in pE199L-expressing cells. Taken together, these data sustain that pE199L induces mitochondrial-dependent apoptosis. 

## 5. Conclusions

In summary, ASFV pE199L was firstly identified as an inducer of apoptosis which competes with BAK for BCL-X_L_ to induce the activation of BAK and BAX and ultimately results in apoptosis (Figure 7). Our findings provide important implications for understanding ASFV pathogenesis. 

## Figures and Tables

**Figure 1 viruses-13-02240-f001:**
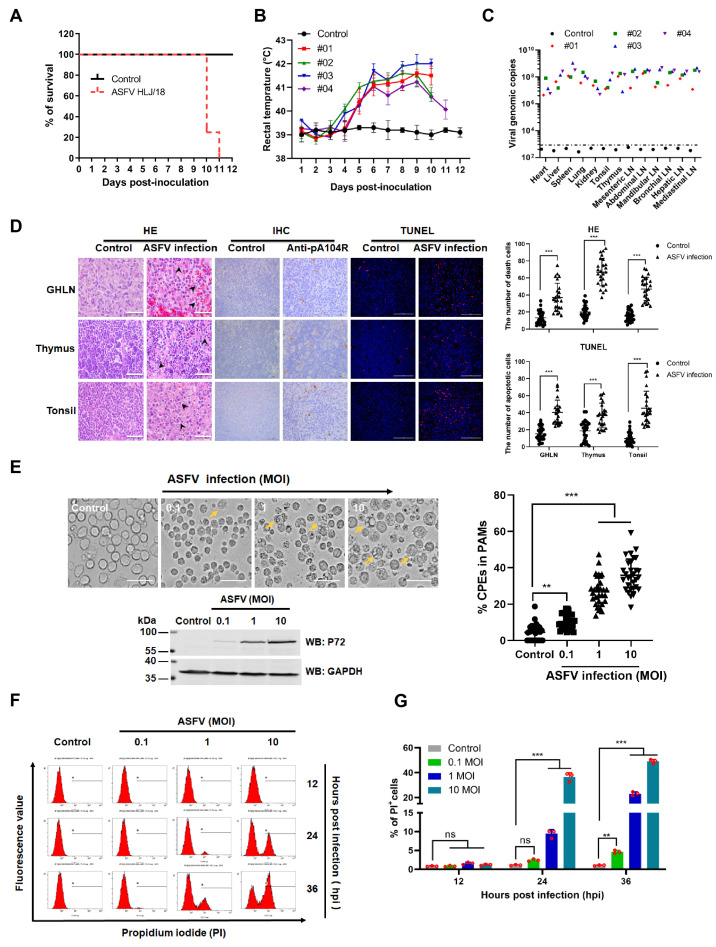
ASFV infection shows tissue injury and cell death in vivo and in vitro. (**A**) Survival rates. (**B**) Rectal temperature. (**C**) Viral DNA of the indicated samples was detected in the dead pigs or surviving pigs that were euthanized at 12 dpi. (**D**) Representative images of tissue sections of gastric hepatic lymph nodes (GHLNs), thymuses, and tonsils from infected or non-infected pigs. Cell death is indicated by black arrowheads in hematoxylin and eosin-stained images (HE) (scale bar: 25 μm). Infected cells (brown) were stained by immunohistochemistry (IHC) with anti-ASFV monoclonal antibody (pA104R) (scale bar: 50 μm) and apoptotic cells were labeled with TUNEL (red) (scale bar: 100 μm). Death or apoptotic cells from 25 fields of 5 random tissue sections were counted under a 40× objective and the results (right) are shown as mean ± SD values, *n* = 25. The significance of the differences between the groups was determined by an unpaired *t*-test with Welch’s test with 2-tails (*** *p* < 0.001). (**E**) Representative images of cytopathic effects (CPEs) caused by ASFV infection in PAMs. PAMs were non-infected or infected with ASFV HLJ/18 at MOIs of 0.1, 1, or 10 for 36 h and then visualized under a microscope (left). Yellow arrows indicate CPEs. Scale bar, 10 µm. The right panels show the statistical results regarding the percentage of CPEs. CPEs from 10 fields (approximately 100 cells/field) were counted under a 20× objective and the results shown as mean ± SD values, *n* = 30 from 3 independent experiments. The significance of the differences between the groups was determined by the Mann–Whitney test (nonparametric *t*-test). (** *p* < 0.01, and *** *p* < 0.001). The expression of p72 indicated ASFV infection was detected (lower image). (**F**,**G**) ASFV caused cell death in PAMs. PAMs were infected with ASFV at MOIs of 0.1, 1, and 10. At different time points (12, 24, and 36 h) post infection, cells were harvested, stained with propidium iodide (PI), and analyzed by flow cytometry. Untreated cells were used as the control. The percentage of PI-labeled cells in (**F**) was quantified (**G**). The significance of the differences between the groups was determined by 2-way ANOVA with Tukey (** *p* < 0.01, and *** *p* < 0.001).

**Figure 2 viruses-13-02240-f002:**
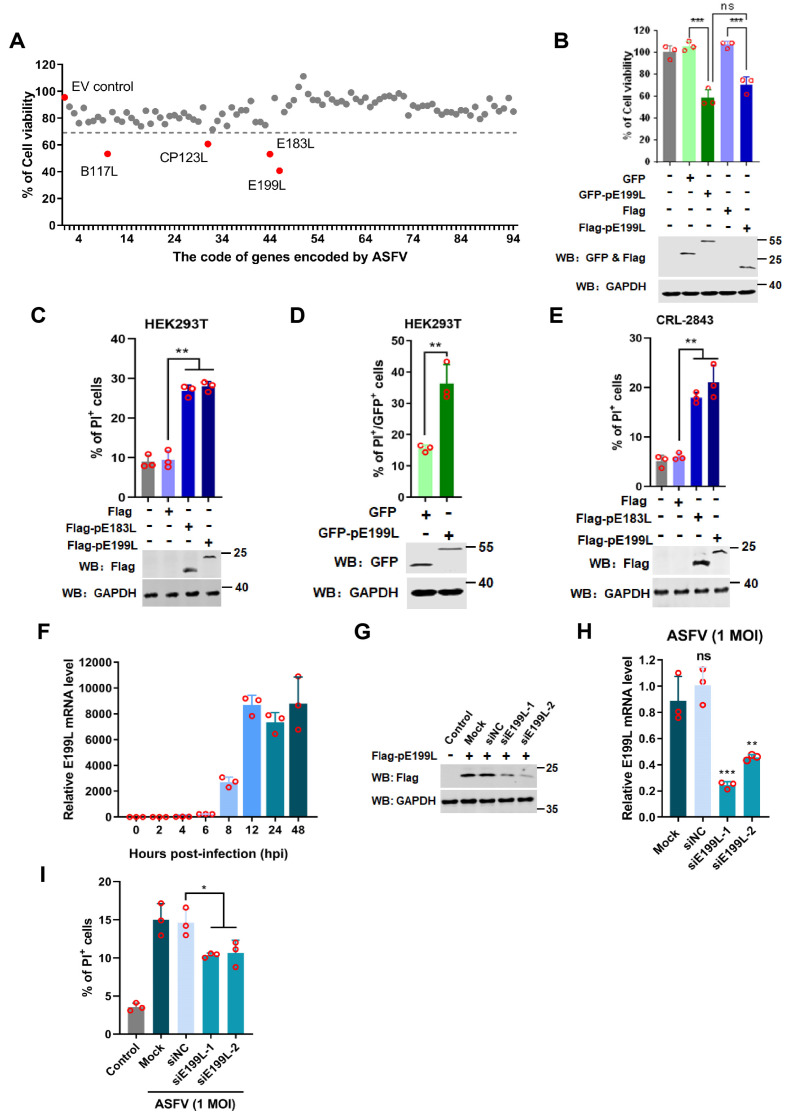
ASFV pE199L induces cell death. (**A**) Screening of the ASFV genes involved in cell death in vitro. HEK293T cells were transfected with a plasmid expressing 1 of the 94 ASFV-encoded proteins for 36 h and then examined for cell viability. The genes corresponding to relative cell viability (normalized to the empty vector control) below the dotted line were considered. (**B**) Detection of the cell viability induced by pE199L. HEK293T cells were transfected with pFlag-E199L or pGFP-E199L for 36 h, followed by ATP activity examination. The HEK293T cells transfected with pCAGGS-Flag (pFlag) or pGFP-C1 (pGFP) were used as control. (**C**,**D**) Detection of cell death induced by pE199L in HEK293T cells. HEK293T cells were transfected with pFlag-E199L, pFlag-E183L as a positive control (**C**), or pGFP-E199L (**D**) for 36 h. The cells were stained with PI, and the percentage of the PI-labeled cells in the total cells (**C**) or in the GFP-expressed cells or the GFP-pE199L-expressed cells (**D**) were analyzed by flow cytometry. (**E**) Detection of cell death induced by pE199L in CRL-2843 cells. CRL-2843 cells were transfected with pFlag-E199L or pFlag-E183L as a positive control for 36 h and then stained with PI. In total, 10,000 cells were analyzed to determine the percentage of PI-labeled cells by flow cytometry. (**F**) Detection of the mRNA level of *E199L* in ASFV-infected PAMs. PAMs were infected with ASFV HLJ/18 at 1 MOI and the mRNA of *E199L* was analyzed at 0, 2, 4, 6, 8, 12, 24, and 48 hpi by qRT-PCR. (**G**,**H**) Testing of siE199L knockdown efficiency. HEK293T cells were transfected with non-targeting siRNA (siNC) or siRNA targeting ASFV *E199L* gene (siE199L-1, siE199L-2) for 12 h followed by transfection with the pFlag-E199L for 24 h and then Flag-pE199L expression testing by Western blotting (**G**). PAMs were transfected with siE199L or siNC for 12 h and then infected with HLJ/18 at 1 MOI for 24 h. The PAMs were subjected to detection of the knockdown effect of ASFV *E199L* by qRT-PCR (**H**). (**I**) Analysis of the effect of pE199L on cell death during ASFV infection. The 10,000 cells in (**H**) were stained with PI and analyzed by flow cytometry. Control: untreated cells; Mock: ASFV-infected cells with non-transfected siRNA. In the Figure 2, + means transfection of indicated plasmids while − means non-transfection of indicated plasmids. The significance of the differences between the groups was determined by the Student’s unpaired *t*-test with 2-tails (parametric test) or the ordinary ANOVA test with Dunnett (* *p* < 0.05, ** *p* < 0.01, and *** *p* < 0.001).

**Figure 3 viruses-13-02240-f003:**
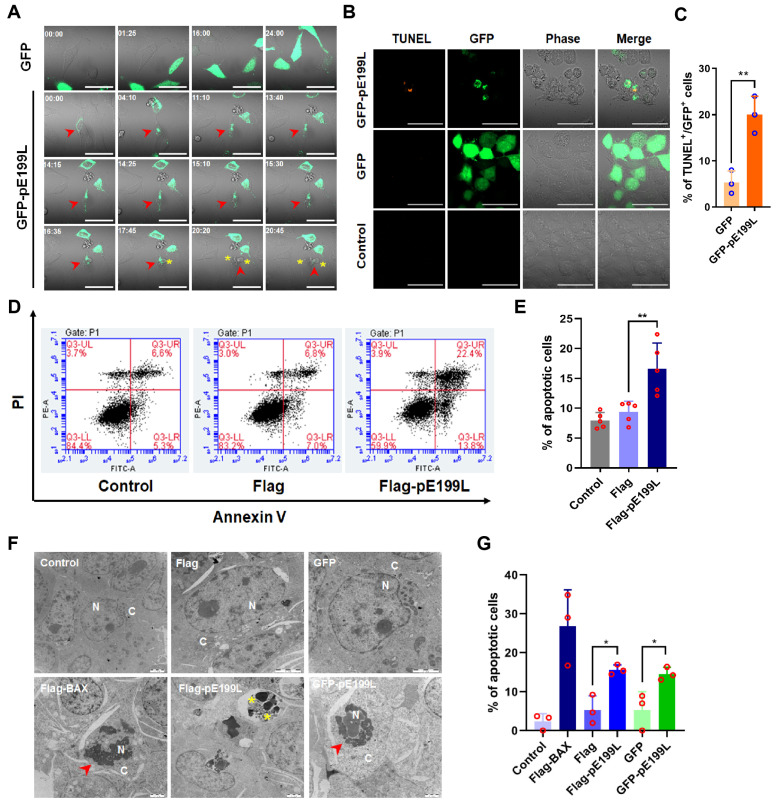
ASFV pE199L induces apoptosis in vitro. (**A**) Representative morphology images of GFP-pE199L-expressing HeLa cells. HeLa cells were transfected with pGFP or pGFP-E199L for 12 h and the morphological changes in 15 random fields were monitored under a 63× oil objective using real-time confocal microscopy for 24 h. Time stands for relative time (hh:mm). Scale bar: 50 μm. Yellow asterisk: formation of vesicles; red arrowhead: apoptotic cells. (**B**) HEK293T cells were transfected with pGFP or pGFP-pE199L plasmids for 36 h and then treated with TUNEL assay as described in Materials and Methods. A representative confocal image illustrated TUNEL-labeled cells (orange) in GFP-pE199L-expressing HEK293T (GFP, green) but not GFP-expressing HEK293T cells. Scale bar: 50 μm. (**C**) Statistical analysis of the percentage of the TUNEL-labeled cells. At least 100 GFP or GFP-pE199L expressing cells in Figure B were counted under a 63× oil objective and the results are shown as mean ± SD values from 3 independent experiments. (**D**,**E**) Analysis of the pE199L-induced apoptosis. HEK293T cells were transfected with pFlag-E199L or pFlag for 36 h, and then stained with Annexin V-FITC and PI. In total, 10,000 cells were analyzed by flow cytometry (**D**). The percentage of apoptotic cells is shown as the mean ± SD values from 5 repetitions in parallel (**E**). (**F**,**G**) Observation of apoptotic cells by transmission electron microscopy (TEM). HEK293T cells were transfected with pGFP-E199L, pGFP, pFlag-E199L, or pFlag for 36 h. The pFlag-BAX was transfected for 36 h as a positive control. The morphology of apoptotic cell was observed by TEM (**F**). N: nucleus; C: cytoplasm; red arrowhead: hyperchromatic nuclear and karyopyknosis; yellow asterisk: apoptotic bodies. Scale bar: 2 μm. Statistical analysis of the percentage of the apoptotic cells (**G**). At least 180 cells from 3 or 4 cell sections were randomly counted and the results are shown as mean ± SD values from 3 independent experiments. The significance of the differences between the groups was determined by the Student’s unpaired *t*-test with 2-tails (parametric test) or the ordinary ANOVA test with Dunnett (* *p* < 0.05, ** *p* < 0.01).

**Figure 4 viruses-13-02240-f004:**
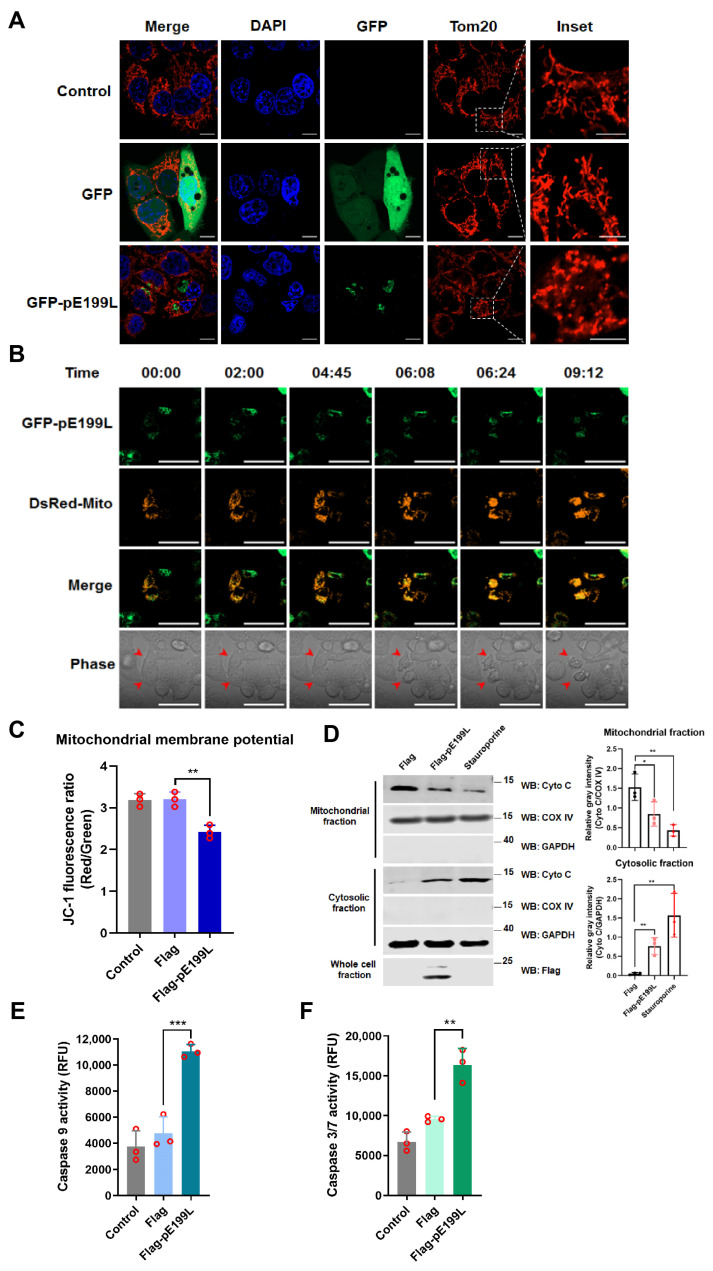
ASFV pE199L triggers the mitochondrial apoptotic pathway. (**A**) Analysis of the morphological changes in mitochondria. HeLa cells were transfected with pGFP-pE199L or GFP for 36 h and were then stained with an antibody against mitochondrial outer membrane 20 (Tom20). The inset images display the typical tubular mitochondrial network in the control and GFP group and the collapsed mitochondria (excessive fission) in GFP-pE199L-expressing cells. Green fluorescence represents GFP or GFP-pE199L while blue indicates nucleus stained by DAPI and red represents mitochondrial stained by Tom20. Inset scale bars: 5 μm; other scale bars: 10 μm. (**B**) Real-time confocal images of HeLa cells co-expressing GFP-pE199L (green) and DsRed-mito (orange). Time stands for relative time (hh:mm). Scale bar: 50 μm. Red arrowhead: apoptotic cell. (**C**) Detection of the mitochondrial membrane potential. HeLa cells were transfected with pFlag-E199L or pFlag for 36 h, stained with JC-1, and then 10,000 cells were quantified by flow cytometry as described in Materials and Methods. (**D**) Western blotting analysis of the translocation of cytochrome C (cyto C) in fractionations from HeLa cells transfected with pFlag-E199L or pFlag or treated with staurosporine as a positive control (left). COX IV and GAPDH were respectively used as the internal controls of the mitochondrial or cytosolic fractions. Ratios of Cyto C and COX IV or GAPDH are shown as the mean ± SD values from 3 independent experiments (right). The gray density of the protein bound was measured by Image J software. (**E**,**F**) Detection of caspase-9 and caspase-3/7 activation. HeLa cells were transfected with pFlag-E199L or pFlag for 36 h and then caspase substrates were added, respectively. The activities of caspase-9 (**E**) and caspase-3/7 (**F**) were measured using an Enspire multiscan spectrum reader. The significance of differences between the groups was determined by an ordinary ANOVA test with Dunnett (* *p* < 0.05, ** *p* < 0.01, *** *p* < 0.001).

**Figure 5 viruses-13-02240-f005:**
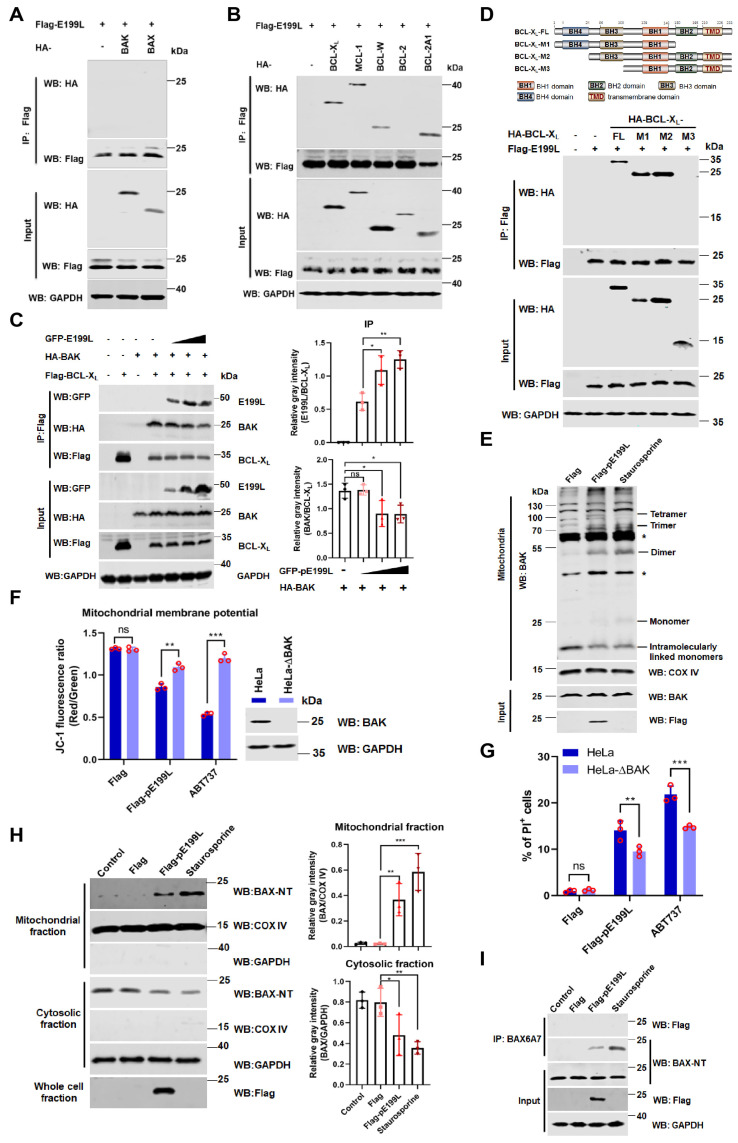
ASFV pE199L promotes BAK activation by competing with BAK for BCL-X_L_. (**A**,**B**) Detection of the interaction of pE199L and apoptotic effectors (BAK and BAX) or anti-apoptotic proteins (BCL-X_L_, MCL-1, BCL-W, BCL-2, and BCL-2A1). HEK293T cells were transfected with pFlag-E199L alone or together with HA-BAK, BAX, BCL-X_L_, MCL-1, BCL-W, BCL-2, or BCL-2A1. Co-IP and Western blotting were performed to test their interactions. (**C**) Disruption of BCL-X_L_-BAK complex by pE199L. HEK293T cells were transfected with Flag-BCL-X_L_, HA-BAK, and pGFP-E199L as indicated, and the competitive interaction between proteins was detected by Co-IP and Western blotting (left). Ratios of pE199L or BAK and BCL-X_L_ are shown as mean ± SD values from 3 independent replicates (right). The gray density of the protein bound was measured by Image J software. (**D**) Analysis of interaction between pE199L and the domain of BCL-X_L_. HEK293T cells were transfected with pFlag-E199L together with HA-BCL-X_L_ and its mutants as indicated. The interactions were detected by Co-IP and Western blotting. (**E**) Induction of BAK homo-oligomerization by pE199L. Mitochondria from HeLa cells transfected with pFlag or pFlag-E199L were cross-linked with BMH as described in Materials and Methods. Asterisk (*) means a nonspecific band. (**F**) Detection of the mitochondrial membrane potential. HeLa or BAK-deficienct HeLa (HeLa-∆BAK) cells were transfected with pFlag-E199L or pFlag for 36 h or treated with ABT737 (10 μM) for 24 h as the positive control, stained with JC-1, and then quantified by flow cytometry as described in Materials and Methods. Western blotting was performed to validate BAK expression in both cell lines. (**G**) Analysis of BAK-dependent apoptosis induced by pE199L. The PI-labeled cells in HeLa and HeLa-∆BAK cells transfected with pFlag-E199L or pFlag were quantified by flow cytometry. The cells treated with ABT737 (10 μM) were used as a positive control. (**H**) Western blotting analysis of the translocation of BAX in the cytosol and mitochondrial fractions from HeLa cells transfected with pFlag-E199L or pFlag or treated with staurosporine as a positive control (left). The COX IV and GAPDH were respectively used as the internal control of mitochondrial or cytosolic fractions. Ratios of BAX and COX IV or GAPDH are shown as the mean ± SD values from 3 independent replicates (right). The gray density of the protein bound was measured by Image J software. (**I**) Western blotting analysis of the BAX activation in HeLa cells transfected with pFlag-E199L or pFlag. Cells were lysed and immunoprecipitated with the anti-BAX6A7 antibody. Equal amounts of the precipitated protein and cell lysates were immunoblotted with an anti-BAX-NT antibody. The cells treated with staurosporine were used as a positive control. The significance of differences between the groups was determined by the ordinary ANOVA test with Dunnett or 2-way ANOVA with Sidak (* *p* < 0.05, ** *p* < 0.01, *** *p* < 0.001).

**Figure 6 viruses-13-02240-f006:**
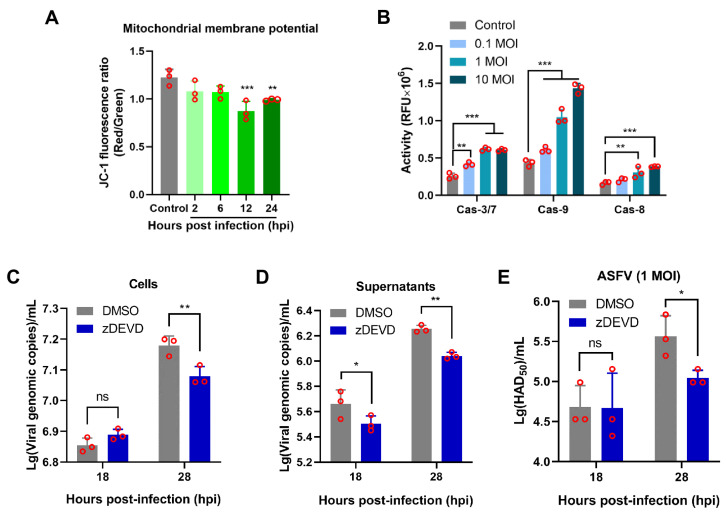
Inhibition of ASFV-induced apoptosis by caspase-3 inhibitor restricted viral growth. (**A**) Analysis of the mitochondrial membrane potential in ASFV-infected PAMs. In total, 10,000 PAMs infected with 1 MOI ASFV HLJ/18 were analyzed at 2, 6, 12, and 24 hpi by flow cytometry as described in Materials and Methods. Non-infected PAMs served as the control. (**B**) Detection of caspase-9, caspase-3/7, and caspase-8 activity in ASFV-infected PAMs. PAMs were infected with ASFV HLJ/18 at an MOI of 0.1, 1, or 10 for 24 hpi, and the activities of caspase-9, caspase-3/7, and caspase-8 were analyzed as described in the Materials and Methods. (**C**–**E**) The effect of apoptosis on ASFV replication was analyzed. PAMs pretreated with z-DEVD-FMK (200 μM) were infected with ASFV HLJ/18 at 1 MOI for 1 h and then cultured with z-DEVD-FMK again for 18 and 28 hpi. The viral growth in the cells (**C**) and supernatants (**D**) was determined with viral p72 gene copies via qPCR or the viral titers in supernatants were detected via the HAD assay (**E**). The significance of the differences between the groups was determined by an ordinary ANOVA test with Dunnett or 2-way ANOVA with Sidak (**p* < 0.05, ** *p* < 0.01, and *** *p* < 0.001).

**Figure 7 viruses-13-02240-f007:**
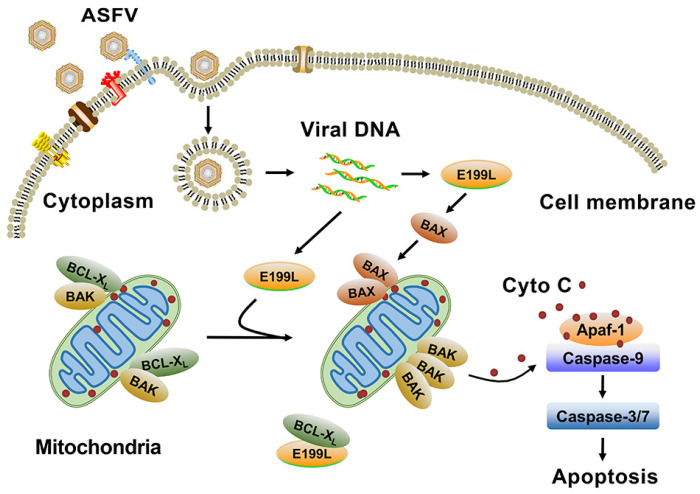
Model of the mitochondrial-dependent apoptosis induced by ASFV pE199L. ASFV replicates predominantly in swine monocytes and macrophages. Following binding to cell surface receptors, the virus particles are first internalized by micropinocytosis and endocytosis. Once internalized, the incoming particles undergo a process of uncoating and genome release. ASFV pE199L is expressed at a late stage of infection. The expressed pE199L can compete with BAK for BCL-X_L_, thus promoting BAK activation. The translocation of BAX to the mitochondria and BAX activation also can be induced by pE199L. These processes can lead to the loss of mitochondrial membrane potential, the release of cyto C, and the activation of caspase-9 and -3/7, ultimately inducing mitochondrial-dependent apoptosis.

**Table 1 viruses-13-02240-t001:** The sequences of the primers used for PCR.

No.	Primer	Sequence
P1	pGFP-E199L	TACAAGTCCGGACTCAGATCTATGTCTTGCATGCCAGTTTCCA
P2		TTATCTAGATCCGGTGGATCCAAAATTGTTTAGGTTTGAAAAAATAAGAG
P3	pCAGGS-HA-E199L	GTTCCAGATTACGCTGAATTCTCTTGCATGCCAGTTTCCACG
P4		ATTAAGATCTGCTAGCTCGAGTTAAAAATTGTTTAGGTTTGAAAAAATAAG
P5	pCAGGS-HA-BCL-X_L_-M1	GTTCCAGATTACGCTGAATTCTCTCAGAGCAACCGGGAGC
P6		ATTAAGATCTGCTAGCTCGAGTCACTCTAGGTGGTCATTCAGGTAA
P7	pCAGGS-HA-BCL-X_L_-M2	GTTCCAGATTACGCTGAATTCAGTCAGTTTAGTGATGTGGAAG
P8		ATTAAGATCTGCTAGCTCGAGTCATTTCCGACTGAAGAGTGAGC
P9	pCAGGS-HA-BCL-X_L_-M3	GTTCCAGATTACGCTGAATTCTACCGGCGGGCATTCAGTGAC
P10		ATTAAGATCTGCTAGCTCGAGTCATTTCCGACTGAAGAGTGAGC
P11	pEGFP-C1-E183L	GTACAAGTCCGGACTCAGATCTGATTCTGAATTTTTTCAACCG
P12		GTTATCTAGATCCGGTGGATCCTTACAAGGAGTTTTCTAGGTC
P13	siRNA-E199L-1	GGAAGACAUCAAACGGUAATT
P14		UUACCGUUUGAUGUCUUCCTT
P15	siRNA-E199L-2	GGUAUAGGUCGGAAAUAUUTT
P16		AAUAUUUCCGACCUAUACCTT
P17	siRNA-NC	UUCUCCGAACGUGUCACGUTT
P18		ACGUGACACGUUCGGAGAATT
P19	qPCR-E199L	GGGCAATATTTCCGACCTATAC
P20		GGGCAACTTATCGTCATTGT
P21	HPRT	GCCGAGGATTTGGAAAAGG
P22		GCACACAGAGGGCTACGATG
P23	ASFV-p72	CTGCTCATGGTATCAATCTTATCGA
P22		GATACCACAAGATCAGCCGT
P23		FAM-CCACGGGAGGAATACCAACCCAGTG-TAMRA

## Data Availability

Data supporting the reported results are available in this article and in the Appendix A.

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
