# Peer review of "African Swine Fever Virus pE199L Induces Mitochondrial-Dependent Apoptosis"

_viruses, 2021, doi:10.3390/v13112240_

Round 1

Reviewer 1 Report

Tingting Li et al characterize the role of pE199L in relation to apoptosis during ASFV infection. They use a range of appropriate assays and a sound analysis, including animal models. The scientific approach is solid and the results are convincing, however in my opinion some controls/statistics are missed and some results could be more discussed.

Reviewer 2 Report

Paper wrote by tingting and co-workers is very interesting and well written. I think it can be considered for publication after minor revisions.

Introduction: citation 5 could be changed with a paper more recente for exaples, Montoya et al., 2021 or Franzoni et al., 2020 or other.

Material and methods: indicate whether the data distribution was assessed. Moreover indicate which test was used. Explain if the analysis performed after is suitable for the distribution highlighted. 

Reviewer 3 Report

This paper describes a role for the gene product of E199L from African swine fever virus, ASFV, in regulating intrinsic apoptosis in infected cells. ASFV, a highly infectious haemorrhagic virus of suids, has a complex genome and relatively few detailed molecular-level investigations on the gene products/proteins have been undertaken. This paper presents an investigation into the role of pE199L hypothesising it interferes with intrinsic apoptosis in the host cell via interaction with the hosts Bcl-2 network. Understanding the role of ASFV genes in regulating the response to ASFV infection is of prime importance in understanding the molecular and cellular mechanisms of action of this deadly virus. The current paper adds to our knowledge on the action of E199L.

One weakness of the current work is that experiments are performed in some non-homologous cell lines (human rather than pig, HEK293T and HeLa cell lines). The underlying assumption is that E199L can target the relevant human proteins or pathways. The assumption needs to be justified as ASFV is not a human pathogen.

Paragraph lines 402-415:

Some sequence analysis of pE199L may be useful. Is there any indication of shared structure or sequence identity with any known proteins?  Does E199L for instance potentially target Bcl-xL through a BH3-motif type interaction? Bcl2A1 does not have a BH3 motif, though it is structurally homologous with Bcl-xL. In addition, key conserved residues of the BH3 motif of Bcl-xL are buried in the folded structure. A pull-down type experiment with a peptide or fragment spanning the BH3 region (of Bcl-xL for instance) alone would be useful as a positive control. The issue with the experiments reported in Fig 5G is that the structural integrity of Bcl-xL is destroyed in the deletion mutants (it could however be an interaction through a linear motif rather than a structural motif, or significant conformational change to Bcl-xL on pE199L interaction occurs. However, these are not proven here).

Paragraph lines 460-474:

Have the authors any comment on a potential role for the BH3-only protein Bid here? Bid spans both the intrinsic and extrinsic pathways and is activated by caspases.

Minor points:

Line 14: ‘disease’ of swine.  Word missing in sentence, add disease

Line 23: ‘the’ mitochondrial …the definitive article missing

Line 61: has ‘been’ reported… Add been

Line 71: replace conduce, possibly ‘advance our understanding’ is more correct.

Line 75: ‘the mechanism of ASFV pathogenesis… Sentence does not make sense

Line 298: ‘decreased remarkably’ I think is probably more correct, (though it might have been expected!).

Line 347: confocal ‘microscopy’. Word missing

Line 543-544: Further investigation into our proposed model was not possible due to the unavailability of antibodies against endogenous pE199L. Remove the words unfortunately and failed, the reagents were not readily available.

Round 2

Reviewer 1 Report

I ha ve no comments. Each point has been revised and authors have provided suitable rebuttals. I would suggest to check discussion section because some phrases sound repetitive. Indeed I would propose to be more clear with infection assays and differences of use between primary PAMs and CRL-2843.